# Circulation of West Nile virus in mosquitoes approximate to the migratory bird stopover in West Coast Malaysia

Jafar Ali Natasha[1], Abd Rahaman Yasmin[1,2]*, Reuben Sunil Kumar Sharma[1], Saulol Hamid Nur-Fazila[3], Md Isa Nur-Mahiza[3], Siti Suri Arshad[3], Hussni Omar Mohammed[4], Kiven Kumar[5], Shih Keng Loong[6], Mohd Kharip Shah Ahmad Khusaini[7]

1 Department of Veterinary Laboratory Diagnosis, Faculty of Veterinary Medicine, Universiti Putra Malaysia, Serdang, Malaysia, 2 Laboratory of Vaccines and Biomolecules, Institute of Bioscience, Universiti Putra Malaysia, Serdang, Malaysia, 3 Department of Veterinary Pathology and Microbiology, Faculty of Veterinary Medicine, Universiti Putra Malaysia, Serdang, Malaysia, 4 Department of Population Medicine and Diagnostic Sciences, Cornell University, Ithaca, New York, United States of America, 5 Department of Pathology, Faculty of Medicine and Health Sciences, Universiti Putra Malaysia, Serdang, Malaysia, 6 Tropical Infectious Diseases Research & Education Centre, Higher Institution Centre of Excellence, University of Malaya, Kuala Lumpur, Malaysia, 7 Department of Conservation of Biodiversity of Wildlife and National Park Malaysia, Kuala Lumpur, Malaysia

* noryasmin@upm.edu.my

## Abstract

Being a tropical country with a conducive environment for mosquitoes, mosquito-borne illnesses such as dengue, chikungunya, lymphatic filariasis, malaria, and Japanese encephalitis are prevalent in Malaysia. Recent studies reported asymptomatic infection of West Nile virus (WNV) in animals and humans, but none of the studies included mosquitoes, except for one report made half a century ago. Considering the scarcity of information, our study sampled mosquitoes near migratory bird stopover wetland areas of West Coast Malaysia located in the Kuala Gula Bird Sanctuary and Kapar Energy Venture, during the southward migration period in October 2017 and September 2018. Our previous publication reported that migratory birds were positive for WNV antibody and RNA. Using a nested RT-PCR analysis, WNV RNA was detected in 35 (12.8%) out of 285 mosquito pools consisting of 2,635 mosquitoes, most of which were *Culex* spp. (species). Sanger sequencing and phylogenetic analysis revealed that the sequences grouped within lineage 2 and shared 90.12%–97.01% similarity with sequences found locally as well as those from Africa, Germany, Romania, Italy, and Israel. Evidence of WNV in the mosquitoes substantiates the need for continued surveillance of WNV in Malaysia.

## Author summary

West Nile virus (WNV) is considered to be one of the neglected diseases in Malaysia as most vector-borne febrile illnesses in humans are predominantly diagnosed as dengue-like illnesses. Nevertheless, WNV might be an additional virus causing the mosquito-borne disease in Malaysia based on data from previous studies. Our group previously

**Data Availability Statement:** All relevant data are within the manuscript and its Supporting Information files.

**Funding:** This work is supported by the Ministry of Higher Education Grant, Malaysia – France Bilateral Research Collaboration 2021 (MATCH 2021) with the grant number KPT MATCH/2021/5540495 (ARY) under Project In Vitro Differential Neuropathogenicity of Nonstructural Proteins (NSPs) of West Nile Virus isolated from Southeast Asia and Europe; and Universiti Putra Malaysia Grantmanship - Geran Putra Berimpak (UPM. RMC.800/2/2/4-GPB-9702300) (ARY). The funders had no role in study design, data collection and analysis, decision to publish, or preparation of the manuscript.

**Competing interests:** The authors have declared that no competing interests exist.

identified the presence of WNV RNA and antibodies in the migratory and resident birds, macaques, bats and swine. However, there is no current data that could confirm the presence of WNV in the vector responsible for transmission of WNV to a wide range of hosts. Using molecular analysis, WNV RNA was detected from *Culex (Cx.) tritaeniorhynchus*, *Cx. vishnui*, *Cx. pseudovishnui*, *Cx. gelidus*, *Armigeres subalbatus*, and *Coquillettidia* spp. The sequence of WNV RNA detected in mosquitoes was similar to the WNV lineage recovered from wild birds at the same location. These findings confirm the presence of an enzootic cycle of WNV in Malaysia and the risk for spill-over to humans and other vertebrates.

## Introduction

The abundance of mosquito vectors combined with movement of migrating birds across the globe has exacerbated the dynamics of West Nile virus (WNV) transmission [1]. WNV-infected birds particularly migrating passerine species are known to be a functional dispersal vehicle for the virus [2,3]. Mosquitoes such as *Culex (Cx.)* species acquire WNV from birds during blood feeding and then transmit the virus to naïve native birds and other susceptible vertebrate hosts [4]. Emergence and re-emergence of this zoonotic mosquito-borne virus in human hosts has led to direct implication in the escalation rate of WNV globally [5].

Current global warming phenomena further contributes to the distribution of climate-sensitive mosquito-borne infectious diseases, as increases in average temperature create an ideal environment for mosquito reproduction and survival [6]. Malaysia, a tropical country with hot and humid weather together with inconsistent rainfall patterns, has an abundance of mosquito species along with high prevalence of mosquito-borne diseases such as endemic dengue and sporadic Japanese encephalitis and chikungunya outbreaks [7–10]. Thus far, there have been no clinical cases of WNV in humans or animals reported in the region. Differential diagnosis for febrile-like illness in humans has always been associated with dengue fever and occasionally Japanese encephalitis if coupled with encephalitis syndrome. WNV, however, seems to be neglected leaving the actual prevalence of the infection is unknown.

Historically, the first detection of WNV in Malaysia was from Sarawak, a Borneo state located in West Malaysia, during 1960 when it was isolated from *Cx. pseudovishnui* [11]. In addition, WNV RNA and/or antibodies were discovered in multiple vertebrate species, including companion birds, water birds, migratory birds, swine, bats, and macaques as well as in humans [12–15]. However, the current status of WNV in mosquitoes in Peninsular Malaysia is lacking. Identifying possible vector species at sites that have a greater risk of WNV transmission is essential to plan for the prevention of WNV outbreaks in future.

The current study was carried out in Kuala Gula Bird Sanctuary, Perak State, and Kapar Energy Venture, Selangor State. Both locations are situated near the Straits of Malacca from where WNV RNA and antibodies previously were detected in migratory and resident water birds [12]. Some of the mosquitoes reported here were collected simultaneously with that previous avian study; however, additional time was needed for the taxonomic identification of the specimens before virus testing, so the data were not presented concurrently. The discovery of WNV in mosquito populations at these locations is a critical step to understand WNV circulation dynamics in the region. Thus, the purpose of this study was to detect the WNV RNA within mosquito populations found near wild bird habitats and to determine the WNV minimum infection rates (MIR). We hypothesized that WNV is presence in the mosquitoes that

have been captured in the sampling area, as the location is where wild birds are available (enzootic cycle).

## Methods

### Ethics statement

Study approval from Institutional Animal Care and Use Committee (IACUC) was received from Universiti Putra Malaysia under reference number UPM/IACUC/AUP NO: R043/17. The approval from the Department of Wildlife and National Parks (DWNP), Peninsular Malaysia, was granted, with the research permit number JPHL&TN(IP):100-6/1/14 to collect mosquitoes from the wild bird landing areas.

### Study design and sampling location

Mosquitoes were collected from two wetland areas located at the Kuala Gula Bird Sanctuary in Perak state (4.94˚ N latitude and 100.49˚ E longitude) and Kapar Energy Venture in the Selangor State (3.12˚ N latitude and 101.32˚ E longitude) (Fig 1).

Both sites face the Straits of Malacca, a frequent stopover spot for migratory birds in Malaysia [16]. Kuala Gula Bird Sanctuary supports a large area of mangrove swamps and river estuary making it a renowned fishing village which rich with small fish, mud crabs and prawns [12,17]. Kapar Energy Venture thermal plant is located on intertidal mudflats containing an ash pond and is located around 3 to 5 km away from the nearest residential area and fishing pier [16]. Routine ringing (leg banding) of migrant birds was regularly conducted by the staff of the Department of Wildlife and National Parks (DWNP), and the sampling of the mosquitoes in this study were conducted in collaboration with the DWNP. Mosquitoes were collected with two sets of traps operated for two consecutive nights from 17 to 19 October 2017 at Kuala Gula Bird Sanctuary and from 12 to 14 September 2018 at Kapar Energy Venture. Both locations support large numbers of migratory birds and provide suitable mosquito habitat. During sampling, both sites experienced monsoon season weather patterns when the rainfall was higher than normal period coupled with thunderstorms as alerted by the official website of Malaysian Meteorological Department.

### Mosquito collection

Two traps were deployed approximately 200 to 500 meters from wild bird mist net locations and resident bird roosting and breeding areas. Temperatures downloaded from the Malaysian Meteorological Department (https://www.met.gov.my/) during mosquito collections were 29.2˚C and 22.0˚C in Kuala Gula and Kapar, respectively [7]. Two Miniature CDC Light Traps Model 512 with a UV light (John W. Hock Company, USA) were suspended at 1.5 m height and baited with dry ice (Linde, Malaysia). The traps were operated overnight from 1800h to 0700h. The collecting bags containing mosquitoes were placed in an icebox and covered with a damp cloth as suggested by Coleman et al. [18] to keep the mosquitoes alive. Upon reaching the laboratory, the collecting bags were stored at -80˚C (Sanyo, Japan) at the Laboratory of Virology, Faculty of Veterinary Medicine, Universiti Putra Malaysia, for later mosquito identification and analysis.

### Mosquito identification and analysis

**Mosquito identification.** Mosquito identification was done on a chill table using taxonomic keys and morphological characteristics as described by Jeffery et al. [19] and Jourdain et al. [20]. Once identified mosquitoes were pooled according to species, sex, and collection

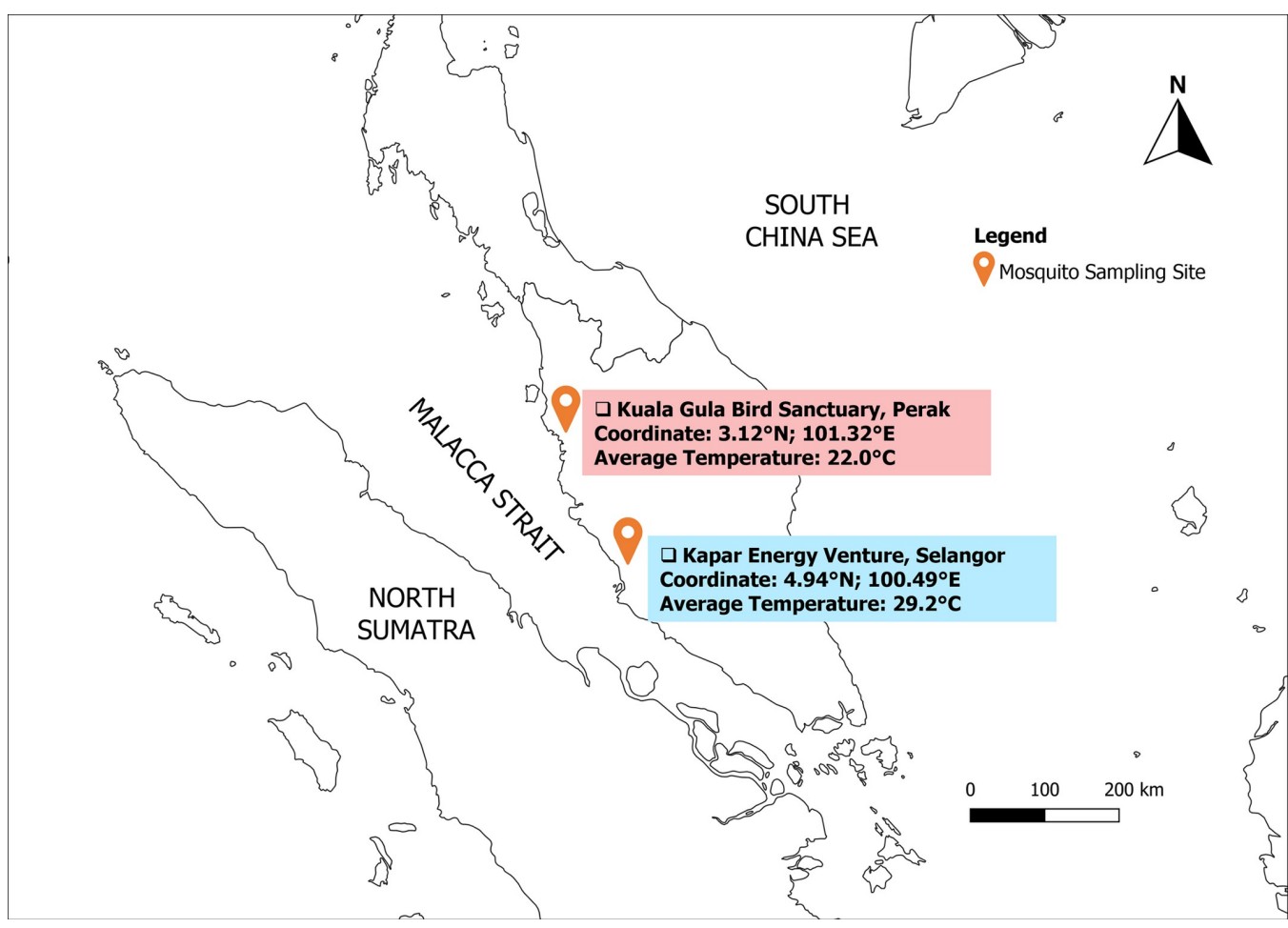

**Fig 1. Sampling sites for mosquito collection.** The wild bird's migration sites and stopover in Malaysia used as sampling location of mosquitoes in this study [marked with orange pin]. The study location map was prepared using Natural Earth website: https://www.naturalearthdata.com/downloads/.

site in a range of one to ten (1–10) mosquitoes per pool as suggested by Patsoula et al. [21]. Maximum pool size of 10 was considered the relevant number to screen for mosquito borne viruses [22]. Pools were stored at –80˚C (Sanyo, Japan) in the Laboratory of Virology, Faculty of Veterinary Medicine, Universiti Putra Malaysia for subsequent RNA extraction.

**Mosquito homogenization.** After adding 500 uL of phosphate buffer saline (PBS), mosquito pools were homogenised at $805 \times g$ using a battery-operated grinder (Sigma, USA) with an autoclavable plastic pestle (Biobasic, Canada) in a container filled with crushed ice until the mosquitoes were fully macerated. All homogenization was done in a Class 2 Biosafety Cabinet (ESCO, Singapore).

**Nested Reverse Transcriptase–Polymerase Chain Reaction (Nested RT-PCR).** RNA was extracted from the homogenised pools using Trizol reagent (Bioline, UK), according to the manufacturer's protocol. A nested reverse transcriptase-polymerase chain reaction (nested RT-PCR) assay was conducted using a MyTaq One-Step RT-PCR kit (Bioline, UK) and MyTaq Red Mix Kit (Bioline, UK) for first and second round PCRs to detect WNV. Two sets of primers targeting the capsid and pre-membrane gene regions were used (Table 1). A synthetic plasmid containing the WNV capsid and pre-membrane regions was used as the positive control [12].

**Table 1. Sequences of primers used in this study.**

|  | Primer | Sequence (5'– 3') | Amplicon Size (bp) | Reference |
|---|---|---|---|---|
| First Round | Forward | CCAATACGTTTCGTGTTGG | 437 | [12] |
|  | Reverse | GGAAATGACCCTGAAGACAT |  |  |
| Second Round[a] | Forward | GCTGGATCGATGGAGAGGTG | 114 | this study |
|  | Reverse | CGGCGGAGCTCAAAACAAAA |  |  |

[a]The second-round primers were designed based on the reference strain WNV-1/Culex/USA/37030146/2003 (Accession no. KX547617) conserved region from Lineage 1 and 2.

The second-round primer set was designed using the NCBI primer designing tool (https://www.ncbi.nlm.nih.gov/tools/primer-blast/) based on sequences from the WNV strain WNV-1/Culex/USA/37030146/2003 (Accession no. KX547617). A pair of primers were designed to flank the RT-PCR product and were amplified using primers previously reported by Ain-Najwa et al. [12], resulting in a PCR product with the expected amplicon size of 114 bp. The first-round protocol was performed under the following cycling conditions: reverse transcription at 45°C for 20 min, polymerase activation at 95°C for 1 min, 40 cycles of denaturation at 95°C for 10 s with annealing at 52°C for 10 s, and extension at 72°C for 30 s, which proceeded to the final extension at 72°C for 5 min. The second round PCR was as follows: initial denaturation at 95°C for 1 min, 30 cycles of denaturation at 95°C for 15 s, annealing at 52°C for 15 s and extension at 72°C for 10 s.

**Phylogenetic analysis.** All positive amplicons from the nested RT-PCR assay were subjected to DNA sequencing. The nucleotide sequences (S1 Table) of the WNV-positive samples were subjected to Basic Local Alignment Search Tool (BLAST) analyses to determine the similarity of the amplicons to WNV strains from other regions. Multiple Sequence Alignment (MAFFT) software version 7 was used to align 52 WNV reference sequences (S2 Table) and the positive sequences from this study and representative WNV sequences from wild birds from the same study sites. The phylogenetic tree was constructed using the maximum likelihood method employing the Jukes-Cantor model in the Molecular Evolutionary Genetics Analysis (MEGA) 7 software. A bootstrapped confidence interval of 1,000 replicates was set. The Newick file was created, and the tree was viewed and edited using the FigTree.v1.4.4.

**Minimum infection rate.** Minimum Infection Rate (MIR) was calculated using the equation below [23]:

$$Minimum\ Infection\ Rate\ (MIR) = \frac{Number\ of\ positive\ pools}{Total\ number\ of\ mosquitoes\ tested} \times 1000$$

The calculation of MIR was further computed by using PooledInfRate software, Version 4.0 [24] to acquire upper and lower limit at 95% confidence interval. The MIR calculation was proceeded only if the total number of mosquito (the denominator) is 1 000 at minimum [25]. The MIR was further evaluated based on the three levels of viral activity as suggested by Sule and Oluwayelu [26] which were the viral activity is nil when the MIR is equal to zero which ultimately concludes as from little to no WNV infection risk on human (Low MIR). Whereas the viral activity may present when the MIR value was between 0.1 and 3.9 which implied the mosquito testing and more vector surveillance (Medum MIR). As for MIR value is equal to more than 4.0 represented the level of viral activity are higher indicating approached WNV risk infection on human (High MIR). The value MIR is necessary as monitoring drive for predicting WNV transmission between mosquito-bird cycle communities [27].

## Results

### Mosquito identification

Overall, 2,635 mosquitoes comprising 11 genera and 19 species were collected (Table 2). Of these, 2,621 were females. Two *Culex* subgenera were identified, namely, *Lophoceraomyia* and *Culex* which included the reputed arbovirus vectors *Cx. tritaeniorhynchus, Cx. vishnui, Cx. pseudovishnui, Cx. quinquefasciatus*, and *Cx. gelidus*.

**Table 2. Mosquito Distribution According to Location, Genus, Species, Sex and Pool Number.**

| Habitat | Genus | Species | Sex M | Sex F | Total Mosquito by Species | Number of mosquitoes per pool | Number of pools | Positive WNV RNA pools | Accession Number |
|---|---|---|---|---|---|---|---|---|---|
| Kuala Gula Bird Sanctuary, Perak | *Aedeomyia (Aedeomyia)* | *catastica* | | 1 | 1 | 1 | 1 | - | |
| | *Aedes (Aedes)* | *albopictus* | | 6 | 6 | 6 | 1 | - | |
| | *Armigeres (Armigeres)* | *kesseli* | | 1 | 1 | 1 | 1 | - | |
| | *Armigeres (Armigeres)* | *maximus* | | 1 | 1 | 1 | 1 | - | |
| | *Culex (Culex)* | *gelidus* | | 36 | 36 | 10 | 4 | - | |
| | | *pseudovishnui* | 4 | 84 | 88 | 10 | 10 | - | |
| | | *tritaeniorhynchus* | 2 | 994 | 996 | 10 | 101 | 1 | ON809701 |
| | | *quinquefasciatus* | 2 | 2 | 4 | 2 | 2 | - | |
| | | *vishnui* | 5 | 274 | 279 | 10 | 29 | - | |
| | *Culex (Lophoceraomyia)* | *spp.* | 1 | 1 | 2 | 1 | 2 | - | |
| | *Mansonia (Mansonioides)* | *annulifera* | | 1 | 1 | 1 | 1 | - | |
| | *Uranotaenia* | *spp.* | | 1 | 1 | 1 | 1 | - | |
| | *Verralina (Verralina)* | *butleri* | | 1 | 1 | 1 | 1 | - | |
| Kapar Energy Venture, Selangor | *Aedes (Aedes)* | *albopictus* | | 2 | 2 | 2 | 1 | - | |
| | *Anopheles (Cellia)* | *spp.* | | 1 | 1 | 1 | 1 | - | |
| | *Armigeres (Armigeres)* | *subalbatus* | | 2 | 2 | 2 | 1 | 1 | TBR |
| | *Coquillettidia (Coquillettidia)* | spp. | | 1 | 1 | 1 | 1 | 1 | ON815484 |
| | *Culex (Culex)* | *gelidus* | | 3 | 3 | 3 | 1 | 1 | ON815483 |
| | | *pseudovishnui* | | 154 | 154 | 10 | 16 | 2 | ON815482 |
| | | *tritaeniorhynchus* | | 550 | 550 | 10 | 55 | 23 | OP580528; OQ383419 |
| | | *vishnui* | | 480 | 480 | 10 | 48 | 6 | OM945733; ON809702 |
| | *Culex (Lophoceraomyia* | *spp.* | | 1 | 1 | 1 | 1 | - | |
| | | *spp.* | | 18 | 18 | 10 | 2 | - | |
| | *Heizmannia (Heizmannia)* | *spp.* | | 1 | 1 | 1 | 1 | - | |
| | *Topomyia* | *spp.* | | 1 | 1 | 1 | 1 | - | |
| | *Verralina (Verralina)* | *butleri* | | 4 | 4 | 4 | 1 | - | |
| Total | | | 14 | 2621 | 2635 | | 285 | 35 | |

M is indicated as male mosquitoes while F is indicated as female mosquitoes. TBR is indicated as to be retrieved. Maximum number of pools per species is 10 while the minimum is 1.

*The species could not be further identified due to damage on required morphological features of specimenl

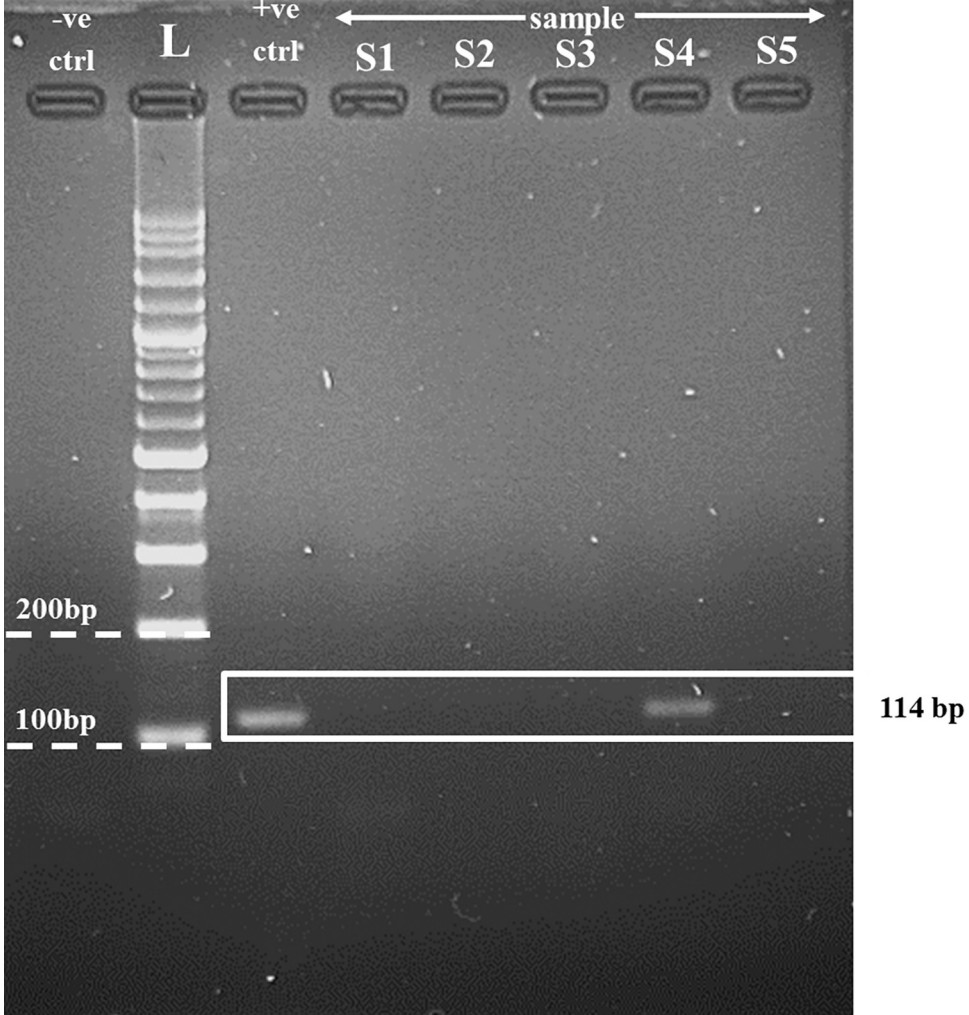

**Fig 2. Bands of positive mosquito samples from the gel electrophoresis procedure.** The expected target size is 114 bp, targeting capsid and pre-membrane genes of the WNV. The gel was prepared with 1.5% agarose gel electrophoresis (w/v). Key: L: 100 bp size ladder; +ve ctrl: negative control (deionised distilled water); +ve ctrl: positive control (WNV plasmid); S1–S5: pooled samples of female *Cx. vishnui* mosquitoes.

## Detection of WNV RNA

WNV RNA was detected in 35 out of 285 (12.28%) mosquito pools (Table 2 and Fig 2). Positive WNV pools were detected from six different species: *Cx. tritaeniorhynchus* (24 pools), *Cx. vishnui* (6 pools), *Cx. pseudovishnui* (2 pools), *Cx. gelidus* (1 pool), *Armigeres subalbatus* (1 pool), and *Coquillettidia* spp. (Table 2). All the positive mosquito pools were comprised of female mosquitoes. One of 24 positive mosquito pools of *Cx. tritaeniorhynchus* was found in Kuala Gula Bird Sanctuary, Perak, while the remaining 23 positive pools were found in Kapar Energy Venture, Selangor.

## DNA Sequencing

**Partial DNA Sequencing and BLAST.** Partial sequences (forward and reverse) of positive amplicons from this study were combined using the BioEdit software (version 7.2.5.0) after generating the reverse complement strands. Positive WNV RNA pools were confirmed by

partial DNA sequencing analysis using BLAST tools. The DNA from viruses in this study had a high similarity with Malaysian strains from Perak and Selangor (wild bird), ranging between 97.01% and 92.01%, and also agreeing with sequences from South African strains with 92.59%; Germany (MH986055), Romania (KJ934710), Uganda (KY523178), and Israel (KC131128) strains with 91.36% homology; and Italian strains (MN939557 and MN939564) with 90.12% homology.

**Phylogenetic analysis.**   A phylogenetic tree constructed using the maximum likelihood method with the Jukes-Cantor model showed evolutionary relationships among strains of WNV from this study. The positive DNA sequences grouped with strains from WNV lineage 2 (Fig 3). All positive amplicons from this study share the same branch, which diverged from other lineages by 2 nucleic acids.

## Minimum Infection Rate

Overall, the minimum infection rate (MIR) was 13.3 (lower and upper limit value showed 8.91 and 17.65 respectively at 95% confidence interval). The value of MIR was denoted as high (greater than 4.0) which indicates possible risk of WNV transmission in the area. MIR calculation in different species of mosquito was not proceeded since number of captured was less than 1000.

## Discussion

Along the west coast of Peninsular Malaysia, Kuala Gula Bird Sanctuary and Kapar Energy Venture are the typical areas for migratory bird stopover along the East Asian-Australasian Flyway [16]. This area also supports a wide variety of water birds that reputedly are the principal reservoirs of WNV and JEV [28,29]. Ain-Najwa et al. [12], who conducted the sampling of wild bird sera and oropharyngeal swabs in the same locations as this study, found that WNV antibody was detected in 29/155 birds (18.71%), while WNV RNA was detected from 16/105 birds (15.2%). These findings indicate high levels of virus transmission. CDC Light Trap was used capture adult mosquitoes and has been widely utilized for vector surveillance [30].

Nested RT-PCR was performed to detect WNV RNA in the homogenised mosquitoes by targeting the conserved regions of WNV between the capsid and pre-membrane protein regions. This type of PCR is highly sensitive as opposed to conventional PCR, which previously detected the presence of WNV from lineage 1 and 2 [31–33]. Overall, 35 out of the 285 pools mosquitoes were positive for WNV RNA (12.28%) and denoted as high MIR in this study with value 13.3. Based on the findings from the Nature scientific report by Chakraburti and Smith [34], low sampling trap density increases the potential for error in MIR estimation, and that it does so synergistically with true MIR values [35].

*Cx. tritaeniorhynchus* was yielded as the highest number of positive WNV pools (24/35 pools). Presence of WNV in field collected *Cx. tritaeniorhynchus* from 2014 to 2015 was considered a breakthrough finding in Assam, India with a 0.36 MIR [32]. An experimental infection study in China by Jiang et al. [35] revealed that *Cx. tritaeniorhynchus* was among the competent vectors of WNV. Similarly, Akhter et al. [36] showed that *Cx. tritaeniorhynchus* required the lowest median WNV infective dose as compared to other mosquito species from Pakistan. These results were interesting from a host blood meal preference perspective, because *Cx. tritaeniorhynchus* typically feeds more frequently on large mammals than birds [37,38]. Fall et al. [38] reported that *Cx. tritaeniorhynchus* densities near to pigeon habitat increased dramatically following the arrival of migratory birds in the region and then declined over time. The high infection prevalence of WNV in this mosquito species supported its role as a vector of WNV during migratory bird arrival during the rainy season. Nevertheless, the

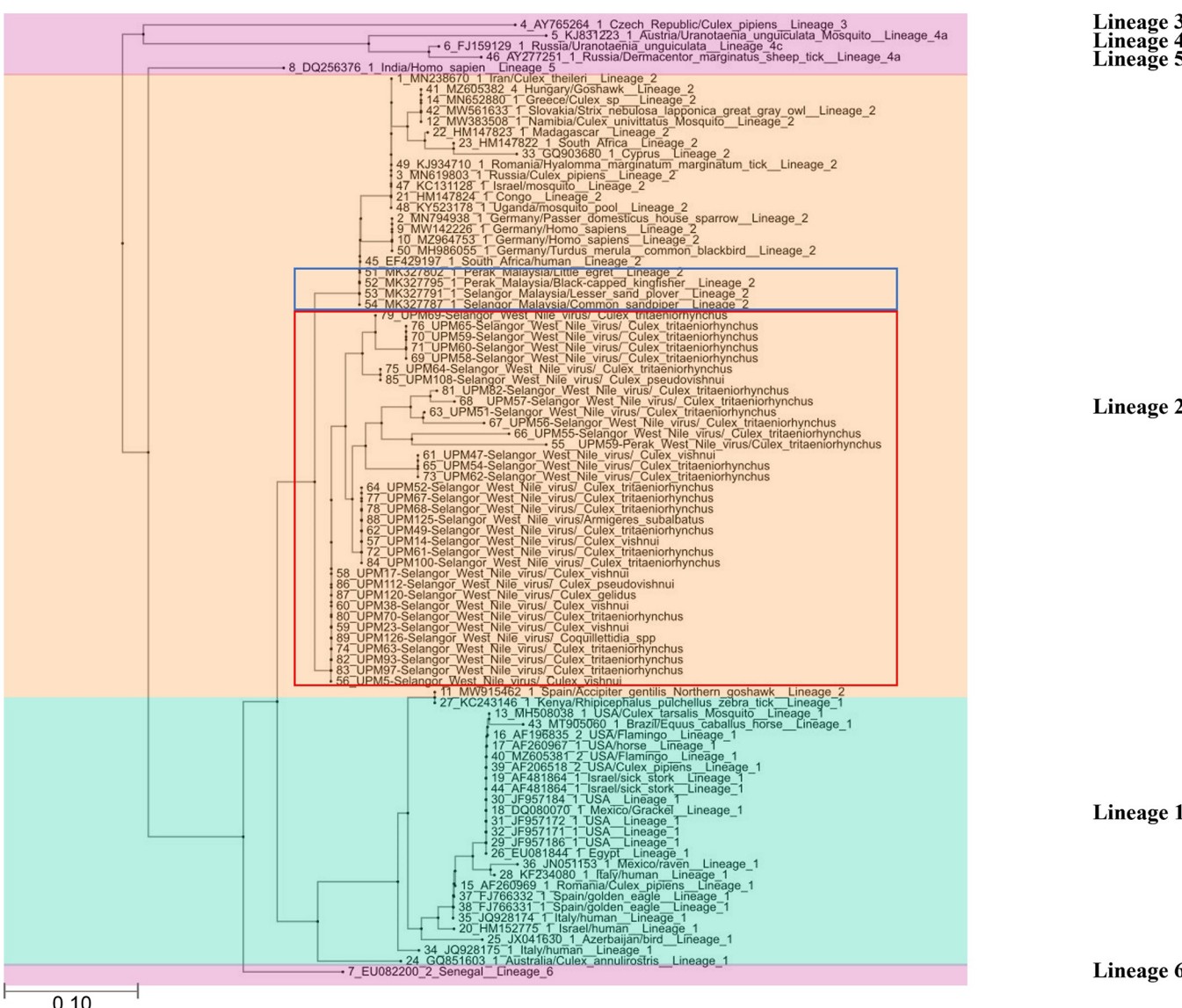

**Fig 3. Phylogenetic analysis of WNV detected in the mosquito pools in Malaysia with other global WNV.** The analysis includes 52 WNV nucleotide sequences from other countries, 4 WNV nucleotides sequences of wild birds from Malaysia (presented in the blue box) and 35 sequences from this study (presented in the red box). The tree branches are colored according to the proposed WNV lineages; green: Lineage 1; orange: Lineage 2; other lineages (3, 4a, 4c, 5 and 6): purple. The tree was constructed using the maximum likelihood method with the Jukes-Cantor model with 1000 bootstrap replicate. Branches are scaled bootstraps.

overall MIR in this study was evidently influenced by the high number of *Cx*. species acquired during sampling due to the type of trap used. Technically, other species such as *Coquillettidia* spp. that have 1/1 WNV positive require further study.

The partial sequencing analysis revealed that the lineage 2 strains detected during our study had approximately 97.01% similarity with WNV from wild birds at the same study site and from South African, Germany, Romania, Italy, and Israel. These studies collectively indicated that WNV lineage 2 has been circulating in Malaysia and implicated migratory birds in viral introduction. The intercontinental relatedness of the WNV isolates from Malaysia may be explained by whole genome sequencing, or at the very least the hypervariable domain.

Some of the sampling of mosquitoes from our study was conducted simultaneously with wild birds sampling in Kuala Gula and Kapar. Our previous data reported that birds from *Ardeidae*, *Charadriidae* and *Scolopacidae* families were shedding WNV RNA from their cloaca and were seropositive to WNV [12]. In addition, JEV antibody also was detected in wild birds in this area [9]. Our study confirmed the presence of an enzootic WNV cycle involving mosquitoes and birds in Peninsular Malaysia. Further studies are warranted to investigate the potential zoonotic link of WNV in the human populations living near the study sites.

## Conclusion

In summary, findings from the current study indicated that WNV was present in most *Cx*. species collected, as demonstrated by the detection of WNV RNA by nested RT-PCR analysis. WNV RNA from mosquitoes and birds originated from the same lineage 2, indicating that this lineage circulates in Malaysia. Nevertheless, our study may be considered preliminary because of the single type of mosquito collection technique, and the time gap between the two study sites. A further entomological investigation is required to describe the blood feeding and oviposition preferences, virus infection rates, and mutations. Programmes targeting the prevention of transmission of WNV through vector control, public awareness, and bio surveillance of pathogens in wild birds are being implemented via a one health initiative among different authorities. This study was in line with Malaysia Strategy for Emerging Diseases and Public Health Emergencies (MYSED) II (2017–2021) under the Ministry of Health, which prioritised enhancing regional capability to rapidly and accurately survey, detect, diagnose, and report outbreaks of pathogens and diseases of public health security concern.

## Supporting information

**S1 Table. Partial DNA Sequence of WNV RNA from Mosquitoes.**
(DOCX)

**S2 Table. List of Reference strain of WNV used in this study.**
(DOCX)

## Acknowledgments

We would like to thank the Department of Wildlife and National Parks (DWNP), Peninsular Malaysia, for the sampling approval. The greatest appreciation is attributed to the late Mr John Jeffery, the writer of entomology books, for his guidance and help with the mosquito identifications.

## Author Contributions

**Conceptualization:** Abd Rahaman Yasmin, Siti Suri Arshad, Hussni Omar Mohammed.

**Data curation:** Abd Rahaman Yasmin.

**Formal analysis:** Jafar Ali Natasha.

**Funding acquisition:** Abd Rahaman Yasmin.

**Investigation:** Jafar Ali Natasha, Mohd Kharip Shah Ahmad Khusaini.

**Methodology:** Jafar Ali Natasha.

**Project administration:** Abd Rahaman Yasmin.

**Resources:** Jafar Ali Natasha, Abd Rahaman Yasmin, Reuben Sunil Kumar Sharma, Saulol Hamid Nur-Fazila, Md Isa Nur-Mahiza.

**Software:** Jafar Ali Natasha.

**Supervision:** Abd Rahaman Yasmin, Reuben Sunil Kumar Sharma, Saulol Hamid Nur-Fazila, Md Isa Nur-Mahiza, Hussni Omar Mohammed.

**Validation:** Abd Rahaman Yasmin, Kiven Kumar.

**Visualization:** Jafar Ali Natasha, Kiven Kumar.

**Writing – original draft:** Jafar Ali Natasha, Abd Rahaman Yasmin.

**Writing – review & editing:** Abd Rahaman Yasmin, Shih Keng Loong.

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
