## [Decision Letter · Decision Letter 0]

26 Jul 2022

Dear Dr Yasmin,

Thank you very much for submitting your manuscript "Circulation of West Nile virus in mosquitoes approximate to the migratory bird stopover in West Coast Malaysia" for consideration at PLOS Neglected Tropical Diseases. As with all papers reviewed by the journal, your manuscript was reviewed by members of the editorial board and by several independent reviewers. In light of the reviews (below this email), we would like to invite the resubmission of a significantly-revised version that takes into account the reviewers' comments. 

We cannot make any decision about publication until we have seen the revised manuscript and your response to the reviewers' comments. Your revised manuscript is also likely to be sent to reviewers for further evaluation.

Sincerely,

Pedro F. C. Vasconcelos

Section Editor

Pedro Vasconcelos

Section Editor

Reviewer's Responses to Questions

**Key Review Criteria Required for Acceptance?**

**Methods**

-Are the objectives of the study clearly articulated with a clear testable hypothesis stated?

-Is the study design appropriate to address the stated objectives?

-Is the population clearly described and appropriate for the hypothesis being tested?

-Is the sample size sufficient to ensure adequate power to address the hypothesis being tested?

-Were correct statistical analysis used to support conclusions?

-Are there concerns about ethical or regulatory requirements being met?

Reviewer #1: Problems:

1. Limited sampling effort. If I read this correctly, mosquitoes were collected at two sites on separate occasions in different years and months using two traps set for two consecutive nights; total effort = 8 trap nights. This is very limited sampling effort and certainly does not lend itself to the extensive analysis of diversity and abundance. These parts of the paper should be deleted. 

2. Mosquito identification. Although this could be do to their presentation, it seemed that mosquitoes were frozen, sorted to sex/genus on a chill table, pooled, refrozen, ground and RNA extracted and tested. Later we learn the specimens were sorted by taxa -- some by species -- before pooling. It was not clear when this was done? The authors included descriptions of the different Culex species that was not necessary as this has been published by various systematists [although the authors did not cite these publications -- e.g., Sirivanakarn 1976.Medical entomology studies III.A revision of the subgenus Culex in the oriental region (Diptera:Culicidae) Contr. Am. Entomol. Inst. 12: 1- 271]. As all specimens were tested for virus, there were no voucher specimens saved for species confirmation and confirmation was not done by molecular methods -- this could be problematic as the 3 abundant Culex species are difficult sort, expecially when rubbed in traps or wet on a chill table. 

3. Virus detection. The authors used a nested RT-PCR to detect virus. Too much detail was given on the methods used which were adopted from previous studies.

Reviewer #2: (No Response)

**Results**

-Does the analysis presented match the analysis plan?

-Are the results clearly and completely presented?

-Are the figures (Tables, Images) of sufficient quality for clarity?

Reviewer #1: Yes.

However, much of this 2was excessive for the data collected and therefore most of the Figures were not necessary.

Reviewer #2: (No Response)

**Conclusions**

-Are the conclusions supported by the data presented?

-Are the limitations of analysis clearly described?

-Do the authors discuss how these data can be helpful to advance our understanding of the topic under study?

-Is public health relevance addressed?

Reviewer #1: Yes

Reviewer #2: (No Response)

**Editorial and Data Presentation Modifications?**

Reviewer #1: The attached manuscript was edited extensively through L214 [see attached file]. In general, the paper was written in a very 'loose fashion' and contained excessive details that should be carefully eliminated where possible. Much of the semantics will require careful scrutiny during revision so terms such as vector, host, reservoir, etc are specific. The paper will need complete revision to reduce length to a short communication. This can be done, in part, by limiting the extensive presentation of the entomological data.

Reviewer #2: (No Response)

**Summary and General Comments**

Reviewer #1: There is limited information on WNV in Malaysia [as pointed out by the authors], which is really the main value of this paper. Their data indicated a very high minimum infection rate in Cx. tritaeniorhynchus from one sampling site on one occasion several years ago. It would be useful to determine if this was a consistent event and why transmission was so high at this one site. 

The paper was too long for the amount of field work done and should be reduced to a short communication focusing on the high infection rates and how this related to data from their previous avian papers. Which avian species had the highest infection rates and were the mosquitoes collected near their nesting/perching sites? Cx. tritaeniorhynchus typically feed more frequently on large mammals than birds [see Reisen and Boreham 1979 AJTMH 28: 408; and other papers], so these results were indeed interesting from that perspective. Perhaps more details about host availability near trap deployment sites would be revealing?

Reviewer #2: This study presents the mosquito trap data associated with a previously published bird infection and seroprevalence study in two natural area study regions in Malaysia. The investigation of WNV and Culex in this region is neglected, so this study contributes data to help fill this gap.

One overall criticism is that very little data are being presented in this study. Several sections, and especially the methods and discussion sections, are excessively large. Normally I would expect this kind of study to be presented as a short communication, but I’m not sure if PLOS NTD has such an article type. Some of the figures and tables are also excessive.

The infection rate is not presented (but should be) but based on the percent of the positive pools, it appears to be very high. The mention that these mosquitoes were collected in October only comes up in the discussion and needs to be presented earlier. The bird and mosquito infection with WNV in this region in October is remarkably high (almost higher than what would be biologically possible for WNV). I would assume the enzootic cycle has seasonality associated with it so did the authors just happen to sample during the peak amplification period in a WNV hotspot? This topic of seasonality deserves more attention.

The authors appear to be fixated on natural wetland areas that have wild migratory birds. However, WNV circulates in urban residential areas in much of the world, facilitated by the presence of Culex mosquitoes and avian amplification hosts (e.g. house sparrows, etc.). Is there a chance the authors are overlooking WNV transmission in other landscapes, including urban, in Malaysia?

The manuscript contains poor English and has inappropriate scientific language. I started making suggestions for specific to the abstract and then stopped providing suggestions for the main body as they were extensive. Given the combination of poor English with poor use of scientific language, extensive edits by a relevant expert are necessary.

Specific comments

Ln 32. ‘mosquito productivity’ would be better than ‘breeding’.

Ln 38. ‘co-occurred’ would be better than ‘coincidently mingled’.

Ln. 39. These study sites could be explained where they are. Malaysia? Coastal?

Ln. 62. Macaque should be plural like the rest.

Ln. 62. Replace ‘prove’ with ‘confirm’.

Ln. 63. Change to: “WNV to a wide range of hosts”.

Ln. 64. ‘WNA’ should be ‘WNV’.

Ln. 67. Could be revised to: “Thus, these findings confirm the presence of an enzootic cycle of WNV in Malaysia and risk for spill-over to humans and animals.”

Ln. 81. Could be changed to: “movement of wild bird amplifications hosts across the globe….”

Ln. 97-99. I assume these study sites are in Malaysia but I have no idea what part of Malaysia.

Ln. 174. What dichotomous key was used to ID the mosquitoes morphologically? With no key mentioned and the heavy emphasis on mosquito communities, with specific species, how confident are the authors in the identifications? Almost appears as though molecular species ID would have helped in this study.

Ln. 221-222. If these primers come from a previous publication, that can be indicated here in the text.

Ln. 230. With WNV circulating around the world for multiple decades and a vast body of research has been conducted along with the development of molecular diagnostic tools, why did this study need to design unpublished primers?

Ln. 114. 114bp is very small and problematic given that this is the sequence used for the phylogenetic analysis.

Ln. 286. I am having trouble finding any details in the methods or results regarding trap effort. How many trap locations were sampled and how many nights?

Ln. 369. You are testing for virus in pooled mosquitoes so it is fine to present the percent of the pools that were infected but you should also consider pool size of those pos pools by calculating the WNV minimum infection rate (usually presented as XX per 1,000 mosquitoes).

Ln. 450-451. You need to be cautious claiming Culex are the dominant mosquito found in these sites given that you only had one sampling tool, and different mosquito species and physiological states are attracted to different trap types.

Ln. 448-459. Actually qPCR would be even more sensitive than nested PCR. I assume you didn’t have a real-time thermocycler, otherwise qPCR would have been advantageous for screening mosquito pools for virus and then a conventional PCR of a larger gene target could have been used for Sanger sequencing.

Ln. 467. All your phylogenetic analyses appear to be based on a very small gene fragment so reporting the percent similarity to other sequences could be mis-leading.

Table 1. If the second round of primers was developed as part of the current study, then that column entry can say ‘this study’ instead of the ‘-‘.

Table 4 could be removed with the data presented in the text.

PLOS authors have the option to publish the peer review history of their article (what does this mean?). If published, this will include your full peer review and any attached files.

Reviewer #1: No

Reviewer #2: No
---

## [Decision Letter · Decision Letter 1]

19 Nov 2022

Dear Dr Yasmin,

Thank you very much for submitting your manuscript "Circulation of West Nile virus in mosquitoes approximate to the migratory bird stopover in West Coast Malaysia" for consideration at PLOS Neglected Tropical Diseases. As with all papers reviewed by the journal, your manuscript was reviewed by members of the editorial board and by several independent reviewers. In light of the reviews (below this email), we would like to invite the resubmission of a significantly-revised version that takes into account the reviewers' comments. 

We cannot make any decision about publication until we have seen the revised manuscript and your response to the reviewers' comments. Your revised manuscript is also likely to be sent to reviewers for further evaluation.

Sincerely,

Pedro F. C. Vasconcelos

Section Editor

Pedro Vasconcelos

Section Editor

Reviewer's Responses to Questions

**Key Review Criteria Required for Acceptance?**

**Methods**

-Are the objectives of the study clearly articulated with a clear testable hypothesis stated?

-Is the study design appropriate to address the stated objectives?

-Is the population clearly described and appropriate for the hypothesis being tested?

-Is the sample size sufficient to ensure adequate power to address the hypothesis being tested?

-Were correct statistical analysis used to support conclusions?

-Are there concerns about ethical or regulatory requirements being met?

Reviewer #1: There was no succinct hypothesis stated or tested in this paper. A previous study detected WNV RNA and antibody in migratory water birds and the current study was done to determine if these birds were able to infect local mosquito populations. The work done established the presence of WNV RNA in mosquitoes collected at the sites of previous bird studies. This indicated that the migrants may have actually been infected locally by mosquitoes??? Mosquitoes were collected by CDC traps -- Supplemental Figure 1 was not necessary. 

Mosquito identification was done on chill tables and the specimen identifications were confirmed morphologically by an expert before homogenization and testing for virus. Detailed descriptions of the species and associated Supplemental Figures 2 and 3 were not necessary. Therefore, there were no voucher specimens saved.

The biggest methodological issue was the species identification of specimens pooled for virus. Line 172 indicated specimens were pooled 'according to genus'? It was unclear then how species specific data was later presented in the results and Table 3????

Reviewer #3: Yes.

Reviewer #4: The objective is not clearly stated. Neither is a testable hypothesis.

The study design is generally appropriate to address the topic based on background information but since the objectives are not clearly stated, it is impossible to state with absolute certainty that the design is very appropriate.

In lines 137-138, the authors don’t sound convincing on the suitability of the sites.

The population is clearly described and appropriate for the methods being described, but the challenge is still the absence of a testable hypothesis.

There is no mention or calculation for an ideal/desired/minimum sample size of mosquitoes to be collected.

From line 251 and Table 3, it’s not clear why the number of pools and number of mosquitoes per pool was chosen.

The generally expected regulatory requirements seem to have been met but it’s unclear from the methods section if ethical approval was sought from an institutional review board. If this were not necessary, then an explanation of the grounds for an exemption would have sufficed.

The statistical analysis methods and results support the conclusions. However, it is difficult to assess the appropriateness of the conclusions because the objectives of the study were not clearly stated.

Reviewer #5: The purpose/objectives of this study are clearly understood and the methods applied are adequate to test the hypothesis of WNV being present in Malaysian mosquitoes that live near migratory bird roosting sites. The mosquito population tested is limited to a small window, but the snapshot style of surveillance does provide a starting point for further investigation and can be used to drum up interest in the subject. The population is able to address the hypothesis tested. The statistical analysis performed fits the data tested, and supports the conclusions drawn by the authors. 

I would suggest the following minor revisions to the methods section:

Line 165: The description of the pools of mosquitoes is lacking. In the Results section (line 258) the authors state that only one pool from 2017 was positive and the remaining positive pools were from 2018. However, the number of total pools from each year is not apparent. This impacts the readers understanding of the situation, as if only 3 pools were collected in 2017, this would mean that 33% pools in 2017 were positive compared to in 2018 where 12% were positive. Whereas, if the pools are evenly split between years, 0.7% (1/142) in 2017 were positive compared to 23.7% (34/143) in 2018. Furthermore, more description could have been provided by stating how many pools were collected at each site for each year. This would have provided an even clearer picture.

Line 181: In the description of the Nested RT-PCR the master mix manufacturer used for the PCR is not listed.

Line 200: The authors state that the RT step of RT-PCR occurred at 95 °C for 20 minutes. I believe this to be a typo as the RT step generally occurs at between 45 and 55 °C and 95 °C is used in the next step (polymerase activation) so as written no change in temperature occurred between these two steps, thus there was no end of the RT step/beginning of the polymerase step.

**Results**

-Does the analysis presented match the analysis plan?

-Are the results clearly and completely presented?

-Are the figures (Tables, Images) of sufficient quality for clarity?

Reviewer #1: Table 3 are the heart of the paper as was Figure 2 showing virus phylogeny. Again, it was unclear how species specific infection rates were determined when specimens were 'pooled according to genus'? This makes calculation of species specific MIRs difficult to understand. In addition, because there were variable numbers of specimens within pools, it would have been better to calculate MLEs which account for variable numbers of specimens tested per pool. 

WNV RNA detected in mosquito pools are not really isolates. Isolates are typically whole viruses preserved in culture or by ultralow freezing.

Reviewer #3: Yes.

Reviewer #4: The analysis presented is in line with the title and the background information, but it's unclear what the analysis plan was from the onset.

Parts of the discussion contain data that should ordinarily be found in the results section. For instance, lines 350-352.

The results are clearly presented. The completeness of the results would be judged better in the presence of a testable hypothesis.

The figures are of sufficient quality for clarity.

Reviewer #5: The authors clearly delineate and address the results from each piece of their analysis plan in the results section and arrange the sections in a manner that follows a logical pattern. Each of the figures was of high enough quality to clearly see the results expressed.

I would suggest the following minor revisions to the Results section:

Line 230 – 248: The mosquito identification section is a bit longer than needed much of the information could have been better summarized in a supplementary table, with examples given in the text.

Line 236: The sentence states that 7 species of the Culex genus were identified, then lists the species specifically. However, the final listed species, simply says Culex spp. Why is this not further elucidated?

**Conclusions**

-Are the conclusions supported by the data presented?

-Are the limitations of analysis clearly described?

-Do the authors discuss how these data can be helpful to advance our understanding of the topic under study?

-Is public health relevance addressed?

Reviewer #1: Conclusions were appropriate.

Reviewer #3: Partially, but I have highlighted the necessary modifications in order to fulfill these requirements in the "Sumary and General comments" box.

Reviewer #4: The first two paragraphs of the Discussion section sound more like background information rather than data supporting (discussing) the findings. 

The larger part of the conclusions is supported by the results presented.

The study/ analysis limitations are omitted.

The discussion on the importance of the data is satisfactory. This includes the public health relevance of the findings.

Reviewer #5: In the discussion the authors appropriately address the limitations of this study, explaining why the methods they used, which admittedly selected for certain mosquito species. The authors do a good job of not over stating the findings of their study. The length of the discussion is a bit long, as they do restate much of what is said in the results, which could be rectified by referencing the results section and briefly summarizing what was written there. They also refer back to the introduction in order to address the public health implications. The authors also discuss how the results of this investigation may lead to further surveillance which I believe was the ultimate goal of the study, as the purpose introduced earlier was to discover if WNV was under surveilled and is more prevalent in Malaysia than previously known. 

I would suggest the following minor revisions in the Discussion and Conclusions sections:

Line 352: The authors state that Culex tritaeniorhynchus had the highest MIR, when technically other species did. Understandably, the authors did not assume that the 1/1 WNV positive Coquillettidia was a realistic MIR, however, this technicality should be addressed, as well as the authors’ reasoning as to why they censored this percentage, and how they addresses the low numbers of several other species. 

Lines 386-395: The authors discuss the environmental conditions of the study sites, but do not discuss the specifics conditions observed. It would be helpful to have conditions laid out, such as cumulated rainfall over the past weeks, and average temperatures (temperature is stated in the study site description in the method, but only the temperature at the time of collection). The authors continue on to reference an article in which temperature affects WNV transmission, but do not discuss how this may have impacted the results of their study. 

Line 309-314: This section could use further clarification, as upon first read the meanings were not explicit.

**Editorial and Data Presentation Modifications?**

Reviewer #1: The paper is too long for the limited amount of sampling done - two traps on two sampling occasions at two locations. It contains too much general information and extraneous details. I have edited the attached file trying to show what areas could be removed/condensed. The data confirms that mosquitoes collected in Malaysia during bird migration were infected with WNV -- period. Species specific data is useful if the pools indeed contained individual species of Culex.

Reviewer #3: (No Response)

Reviewer #4: Minor revision recommended. To include clear objectives, hypotheses, and limitations.

Reviewer #5: (No Response)

**Summary and General Comments**

Reviewer #1: The data shows that females from a several Culex species collected by CDC CO2 trap were infected with WNV L2 and that this RNA was similar to RNA extracted and sequenced from migratory birds collected at the same study sites. This indicates that there is most likely an enzootic cycle established at these wetlands. Recommendations for additional studies was well-taken.

Reviewer #3: The manuscript "Circulation of West Nile virus in mosquitoes approximate to the migratory bird stopover

in West Coast Malaysia" by Natasha et al reports an interesting study that aimed at evaluating the presence of WNV in mosquitoes that were sampled near to a migratory bird stopover. This is an interesting study and the manuscript is well-written. Important changes in the manuscript have been performed according the considerations of the other reviewers. However, I still have some considerations and concerns about the study, as follows:

1. The species of the mosquitoes were not confirmed through a molecular test, which is concerning mainly because of the similarities between Culex species that may compromise the visual identification.

2. Did each mosquito pool comprise only male or female mosquitoes, or the mosquitoes were mixed regardless of sex in an aleatory way? This information should be clear in the manuscript.

3. Lines 401-2: "the involvement of mosquitoes as competent vectors". The fact that the mosquitoes are infected with the virus suggests but does not prove that the mosquitoes from the region are competent vectors to infect vertebrates, since the virus can perpetuate in a mosquito population through other mechanisms such as vertical transmission. Moreover, the vectorial competence varies between mosquitoes of the same species from different geographical areas. Therefore, I suggesting the rephrasing of this excerpt.

4. Lines 412-3: "This study suggests the possible role of Culex species as the main vector of WNV transmission in the region". Again, as I explained above in the point 3, the fact that the mosquitoes of a given species were found to be infected with the virus does not necessarily mean that they are the main vector of the WNV, because the other mosquito species with an inferior MIR could be more competent to transmit the virus through bite.

Furthermore, I suggest that future investigations evaluate mosquito larvae collected in that geographical area for investigating vertical transmission of the virus, which can be a mechanism that is favoring the perpetuation of the WNV in the mosquito populations. Moreover, it would be interesting to investigate the vectorial competence of the mosquitoes from that area, in order to establish the capacity of the mosquitoes from the region to infect vertebrates, since previous studies demonstrate that the transmission of arboviruses varies between mosquitoes from different geographical areas.

Reviewer #4: The manuscript lays a lot of emphasis on the description of the methods and findings, but omits crucial sections on objectives, hypotheses, and limitations.

Reviewer #5: In the introduction:

Line 86: The authors introduce the concepts of Malaysia having topic weather, as well as endemic dengue and sporadic JEV and CHIKV outbreaks. They cite a source for the climate concepts but not for the endemic dengue or outbreaks of mosquito-borne viruses. This would strengthen the argument for the need for surveillance of WNV in mosquitoes.

PLOS authors have the option to publish the peer review history of their article (what does this mean?). If published, this will include your full peer review and any attached files.

Reviewer #1: No

Reviewer #3: No

Reviewer #4: No

Reviewer #5: No
---

## [Decision Letter · Decision Letter 2]

21 Feb 2023

Dear Dr Yasmin,

Thank you very much for submitting your manuscript "Circulation of West Nile virus in mosquitoes approximate to the migratory bird stopover in West Coast Malaysia" for consideration at PLOS Neglected Tropical Diseases. As with all papers reviewed by the journal, your manuscript was reviewed by members of the editorial board and by several independent reviewers. The reviewers appreciated the attention to an important topic. Based on the reviews, we are likely to accept this manuscript for publication, providing that you modify the manuscript according to the review recommendations. 

Sincerely,

Pedro F. C. Vasconcelos

Section Editor

Pedro Vasconcelos

Section Editor

Reviewer's Responses to Questions

**Key Review Criteria Required for Acceptance?**

**Methods**

-Are the objectives of the study clearly articulated with a clear testable hypothesis stated?

-Is the study design appropriate to address the stated objectives?

-Is the population clearly described and appropriate for the hypothesis being tested?

-Is the sample size sufficient to ensure adequate power to address the hypothesis being tested?

-Were correct statistical analysis used to support conclusions?

-Are there concerns about ethical or regulatory requirements being met?

Reviewer #1: The purpose of the study was clear.

Reviewer #5: Introduction: Line 105 - Hypothesis statement has some grammatical issues and reads as if the hypothesis is that wild birds are the bloodmeal source for WNV infected mosquitoes in the sample areas. Given that the paper does not test this, the statement should be reworded to hypothesize the presence of WNV in the mosquitoes of these areas, as that was what was tested.

**Results**

-Does the analysis presented match the analysis plan?

-Are the results clearly and completely presented?

-Are the figures (Tables, Images) of sufficient quality for clarity?

Reviewer #1: The Results are clearly presented.

The authors should carefully recheck their MIR calculations per species? I got different results [see attached manuscript].

Reviewer #5: Line 288: Minor grammatical issue makes sentence harder to read.

**Conclusions**

-Are the conclusions supported by the data presented?

-Are the limitations of analysis clearly described?

-Do the authors discuss how these data can be helpful to advance our understanding of the topic under study?

-Is public health relevance addressed?

Reviewer #1: The conclusions are supported by the data.

Reviewer #5: (No Response)

**Editorial and Data Presentation Modifications?**

Reviewer #1: I downloaded the .pdf version and from this downloaded the cleaned .docx version. I read this version and added edits/comments to improve English, clarity and syntax. Hopefully this will allow the authors to prepare an acceptable final version. 

Semantics: The authors use the terms virus, infection and disease in the same context, which I feel is incorrect. The name of virus should be used when discussing the pathogen, infection occurs after the pathogen is transmitted to the host, and disease describes the response of the host to the infection.

Reviewer #5: (No Response)

**Summary and General Comments**

Reviewer #1: The paper should be acceptable after one more careful revision.

Reviewer #5: (No Response)

PLOS authors have the option to publish the peer review history of their article (what does this mean?). If published, this will include your full peer review and any attached files.

Reviewer #1: No

Reviewer #5: No

Figure Files:

Data Requirements:

Reproducibility:

References

---

## [Editor Report · Decision Letter 3]

21 Mar 2023

Dear Dr Yasmin,

We are pleased to inform you that your manuscript 'Circulation of West Nile virus in mosquitoes approximate to the migratory bird stopover in West Coast Malaysia' has been provisionally accepted for publication in PLOS Neglected Tropical Diseases.

Best regards,

Pedro F. C. Vasconcelos

Section Editor

Pedro Vasconcelos

Section Editor

---

## [Editor Report · Acceptance letter]

3 Apr 2023

Dear Dr Yasmin,

We are delighted to inform you that your manuscript, "Circulation of West Nile virus in mosquitoes approximate to the migratory bird stopover in West Coast Malaysia," has been formally accepted for publication in PLOS Neglected Tropical Diseases.

Best regards,

Shaden Kamhawi

co-Editor-in-Chief

Paul Brindley

co-Editor-in-Chief
